# Intelligent Director: An Automatic Framework for Dynamic Visual Composition using ChatGPT

## Abstract

With the rise of short video platforms represented by TikTok, the trend of users expressing their creativity through photos and videos has increased dramatically. However, ordinary users lack the professional skills to produce high-quality videos using professional creation software. To meet the demand for intelligent and user-friendly video creation tools, we propose the Dynamic Visual Composition (DVC) task, an interesting and challenging task that aims to automatically integrate various media elements based on user requirements and create storytelling videos. We propose an Intelligent Director framework, utilizing LENS to generate descriptions for images and video frames and combining ChatGPT to generate coherent captions while recommending appropriate music names. Then, the best-matched music is obtained through music retrieval. Then, materials such as captions, images, videos, and music are integrated to seamlessly synthesize the video. Finally, we apply AnimeGANv2 for style transfer. We construct UCF101-DVC and Personal Album datasets and verified the effectiveness of our framework in solving DVC through qualitative and quantitative comparisons, along with user studies, demonstrating its substantial potential.

## 1 Introduction

With the booming development of short video platforms such as *TikTok*[1], more and more users around the world use TikTok, and its Daily Active User (DAU) has exceeded one billion. A large number of users take photos and videos, and use video creation technology to upload exquisite works to the short video platform to share their wonderful creativity. Video creation involves the clever combination of captions, images, videos, music, special effects, and other materials to create expressive new videos. However, creating excellent video works requires not only artistic aesthetics but also proficiency in various professional creation software, such as Adobe Premiere and Final Cut. For ordinary users, creating excellent video works is extremely challenging, especially for those who lack creation thinking or are not familiar with creation software. Therefore, the demand for intelligent, user-friendly video creation tools is increasing rapidly. In this paper, we addresses the challenge of enabling ordinary users to create high-quality storytelling videos without the need for professional skills in video creation tools.

As shown in Figure 1(a), we formalize intelligent video creation into a new task called *Dynamic Visual Composition (DVC)*, which refers to automatically integrating multiple media elements such as text, images, videos, and audio based on user instructions or requirements to create storytelling videos. DVC requires well alignment of visual elements with captions and music to effectively convey the story of the creator in video format. Therefore, overall, it is a very interesting and challenging artificial intelligence task. To the best of our knowledge, this extremely complex task has so far been virtually unexplored. DVC can be used not only in entertainment fields such as short video platforms, but also in many fields such as advertising, education, and virtual reality. For example, in digital advertising, DVC can vividly display product features and usage scenarios to better attract the attention of users.

However, dynamic visual composition is a very challenging task that requires multiple capabilities, including 1) *Multi-modal understanding.* Deeply understanding the semantics of text, images, videos, and audio to

---

[1] https://www.tiktok.com/

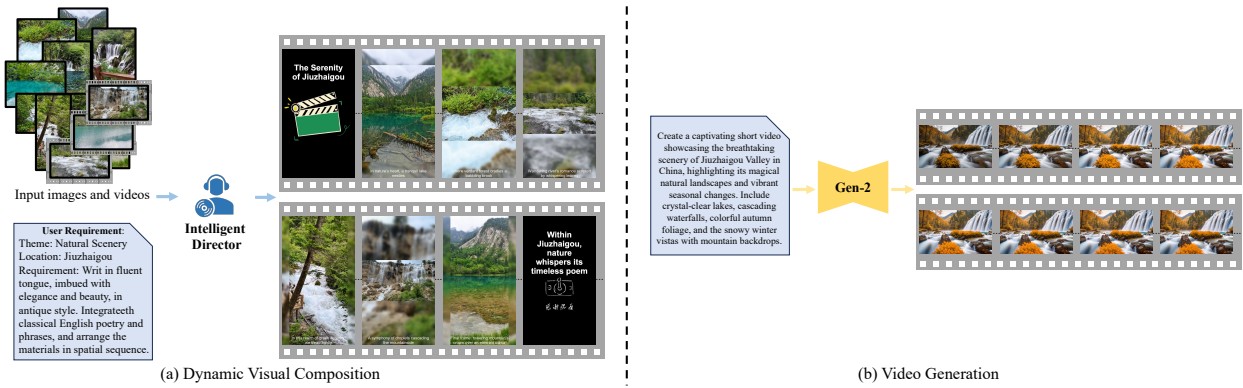

Figure 1: (a) The **Dynamic Visual Composition** (DVC) task, where images, videos, text, and audio are automatically integrated based on user requirements to create a storytelling video using our Intelligent Director. (b) However, traditional **video generation**, like Gen-2, is limited to producing single-scene videos.

extract important information; 2) *Sequencing reasoning.* Reasoning about the relationship between images and videos to form permutations and combinations that build high-quality storylines; (3) *Video Composition.* Create an artistic video by cleverly combining images, videos, text, audio, special effects, etc.

Video generation is a closely related task to DVC, wherein the objective is to produce realistic videos characterized by superior visual quality and consistent motion. As shown in Figure 1(b), the video generation model Gen-2[2] generates a very realistic but single scene landscape video based on the input text, and the single scene is a problem with most current video generation models (Ho et al., 2022a; Singer et al., 2022; Esser et al., 2023). Conversely, as shown in Figure (a), we introduce an intelligent director framework based on ChatGPT that mimics the directorial abilities demonstrated in cinema, an exceedingly complex undertaking that requires the seamless integration of multi-modal understanding, sequencing reasoning, and video composition. The Intelligent Director can automatically integrate input images and videos according to user requirements to synthesize videos with a coherent story.

Our Intelligent Director framework consists of four main steps, namely *Caption Generation*, *Music Retrieval*, *Video Composition*, and *Style Transfer*. In caption generation, we directly employ LENS (Berrios et al., 2023) to generate image descriptions of images. For videos, we first use perceptual hashing (pHash) (Zauner, 2010) to identify video key frames, followed by using LENS to generate descriptions for each key frame. As the descriptions generated by LENS are simplistic and lack consideration of the contextual information from all images and videos, they are insufficient for constructing a storyline. Therefore, we leverage the generation and reasoning capabilities of ChatGPT to generate coherent and storytelling captions and recommend suitable music name. In music retrieval, we use the music names recommended by ChatGPT to search a large music library, obtain the best-matched music to form a high-quality video. In video composition, after gathering all the materials for video creation (including images, videos, captions, and music), we first insert captions into images and videos through caption fusion. Subsequently, we resize images and videos to the target resolution through material fine-tuning. During each transition between materials, a random switching animation is selected. Finally, music fusion is conducted, aligning material switches with the music beats through beat detection and playback time adjustments, ensuring a seamless integration of video and music. Finally, we employ AnimeGANv2 for style transfer, such as adopting the animated style of Kon Satoshi.

Our contributions in this paper are summarized as follows: 1) We propose Dynamic Visual Composition, an interesting and challenging task that involves integrating images, videos, and other materials based on user requirements to create storytelling videos. 2) We introduce Intelligent Director, an automatic framework based on ChatGPT, addressing DVC through four steps: caption generation, music retrieval, video composition, and style transfer. 3) We construct the UCF101-DVC Dataset and Personal Album

---

[2]https://research.runwayml.com/gen2

Dataset for DVC, demonstrating the effectiveness of our framework in solving DVC through qualitative and quantitative comparisons, along with user studies, demonstrating its significant potential.

## 2 Related Work

### 2.1 Video Generation

Recently, AI-Generated Content (AIGC) has received widespread attention. Text-to-video generation based on the diffusion model (Rombach et al., 2022) has greatly improved the video generation effect, so it has received considerable attention (Ho et al., 2022a; Singer et al., 2022; Hong et al., 2022; Khachatryan et al., 2023; Zhou et al., 2022; Rombach et al., 2022; Blattmann et al., 2023; Esser et al., 2023). Notably, a recent trend is that due to the superior generative capabilities of large-scale text-to-image models, many methods attempt to transfer their knowledge and even extend to text-to-video generation (Singer et al., 2022; Hong et al., 2022). Make-a-video (Singer et al., 2022) effectively addresses text-to-video generation by using synthetic data for self-supervision. CogVideo (Hong et al., 2022) is based on the text-to-image model CogView2 (Ding et al., 2022), which adopts a multi-frame rate hierarchical training strategy to facilitate the alignment of text and video. Other works (Ho et al., 2022b; Zhou et al., 2022) utilizing Latent Diffusion Model (LDM) (Rombach et al., 2022) introduced a temporal adjustment technique that enabled high-resolution video generation by introducing effective fine-tuning parameters. On this basis, Text2Video-Zero (T2V-Z) (Khachatryan et al., 2023) introduces an adjustment-free zero-shot video generation method without intensive training or the use of large-scale video data sets. MovieFactory (Zhu et al., 2023) is a work most similar to our framework, which uses ChatGPT to extend user-provided text into detailed scripts, then utilizes a diffusion model to generate video, and finally obtain audio through retrieval models to achieve automatic movie creation. Although the aforementioned works have made significant progress in generating high-fidelity videos, it is worth noting that DVC emphasizes the automatic fusion of multiple media elements to create videos, while video generation aims to generate realistic videos. Therefore, existing video generation works cannot effectively solve the challenge of DVC.

### 2.2 Large Language Models

Large language models (LLMs), characterized by their emergent abilities (Wei et al., 2022) and proven effectiveness across diverse complex tasks such as decision-making (Li et al., 2022), program synthesis (Austin et al., 2021), and prompt engineering (Zhou et al., 2022), are broadly classified into two categories: open source models, including LLaMA (Touvron et al., 2023), PaLM (Chowdhery et al., 2023), etc., and closed source models represented by the GPT series (Radford et al., 2018; 2019; Brown et al., 2020; Ouyang et al., 2022; Achiam et al., 2023). LLMs also perform well on tasks in other modalities (Saharia et al., 2022; Koizumi et al., 2020; Chen et al., 2023; Brooks et al., 2023) (i.e., audio, video, and image). Recently, there have been efforts to integrate LLMs with diffusion models (Rombach et al., 2022), leveraging prompts generated by LLMs to produce more reliable results. DirecT2V (Hong et al., 2023) utilizes LLMs to divide user inputs into separate prompts for each frame, generating frame-by-frame descriptions. These descriptions then guide diffusion model in video generation. Free-Bloom (Huang et al., 2023) employs LLMs to generate a semantic coherence prompt sequence and LDM to generate the high-fidelity frames. These methods combine the text processing capabilities of LLMs with the text-to-image generation capabilities of stable diffusion. However, relying solely on the text processing capabilities of LLMs cannot effectively solve DVC. In this paper, we successfully combine ChatGPT with multi-modal understanding and video composition capabilities to effectively solve DVC.

## 3 Overview

### 3.1 Dynamic Visual Composition

In this paper, we introduce a new task called dynamic visual composition, which can automatically create storytelling videos by dynamically combining images and videos based on user requirements. Given a set of

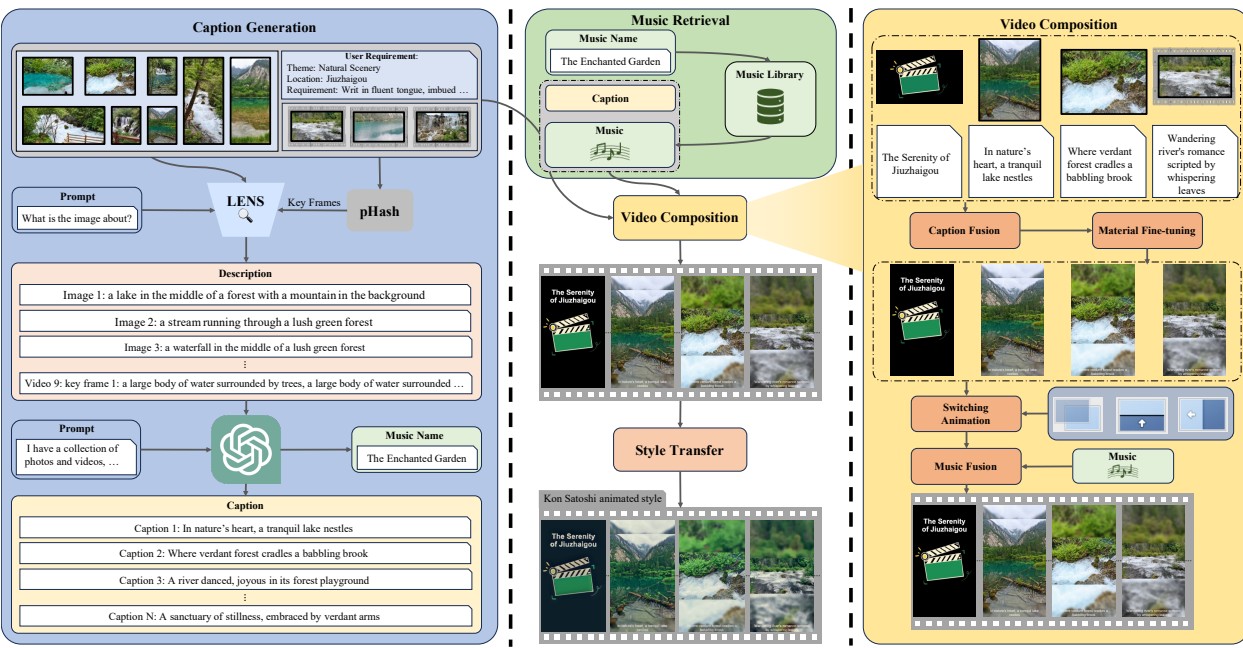

Figure 2: **An overview of our Intelligent Director framework for Dynamic Visual Composition.** Intelligent Director consists of four main steps: (1) Caption Generation, (2) Music Retrieval, (3) Video Composition, and (4) Style Transfer. In caption generation, LENS generates descriptions for images and video key frames extracted by pHash, and then ChatGPT creates coherent and storytelling captions and recommends a suitable music name. In music retrieval, we utilize the music name recommended by ChatGPT to search a large music library for the best-matched music. In video composition, the seamless integration of captions, images, videos, and music is achieved through four steps: caption fusion, material fine-tuning, switching animation, and music fusion. Finally, in style transfer, AnimeGANv2 transforms the video to other styles, such as the animated style of Kon Satoshi.

input images $I = [I_1, I_2 \ldots I_n]$ and videos $V = [V_1, V_2 \ldots V_m]$, along with user requirements $U$, the goal is to create a visually and temporally coherent video $\hat{V}$ that adheres to the specified composition requirements.

DVC goes beyond a mere combination of images and videos. It requires proficiency in different fields such as image and video understanding, text understanding, image captioning, music integration, style transfer, etc. The main challenge is to build a fully automatic, intelligent, user-friendly framework capable of addressing multi-modal understanding, sequencing reasoning, and video composition simultaneously.

## 3.2 Intelligent Director

In this paper, we introduce Intelligent Director, an automatic framework for DVC. It systematically addresses the complex issues involved in generating visually engaging and temporally coherent video. Users can start the creative process with concise descriptions, simplifying the production of video creation works. Intelligent Director adeptly tackles the challenges of democratizing video creation, providing a user-friendly tool for individuals with varying levels of expertise.

In order to create beautiful and storytelling videos, we simulate the rule-of-thumb that directors use to create videos in practice. Initially, we rearrange and combine all the images and videos, prepare appropriate captions and music, then combine all the materials for video synthesis, and finally change the style of the video (such as animation style).

We present an overview of our Intelligent Director framework for Dynamic Visual Composition in Figure 2, Our framework consists of four main steps: Caption Generation, Music Retrieval, Video Composition, and Style Transfer. However, the core of our framework focuses on Caption Generation and Video Composition.

1. In Caption Generation, we directly use LENS (Berrios et al., 2023) to generate image descriptions for images. As for videos, we initially employ Perceptual Hashing (pHash) (Zauner, 2010) to recognize video key frames, followed by using LENS to generate descriptions for each key frame. Then, we utilize the generation and reasoning capabilities of ChatGPT to generate coherent and story-telling captions, along with recommending a suitable music name.

2. In Music Retrieval, we leverage the music names recommended by ChatGPT to search a large music library, obtaining the best-matched music to ensure a high-quality match with the video.

3. In Video Composition, we begin by integrating captions into both images and videos through caption fusion. Following that, we adjust the scale of images and videos to the target resolution during material fine-tuning. For each transition between materials, a random switching animation is selected. Finally, music fusion is applied to synchronize the material switches with the music beats, ensuring a seamless integration of video and music.

4. In style transfer, we employ AnimeGANv2[3] for style transfer, for instance, mimicking the animated style of Kon Satoshi.

## 4 Method

In this section, we first introduce the generation of captions through LENS and ChatGPT(Section 4.1). Then, getting the best-matched music through music retrieval is introduced (Section 4.2). Then we introduce the four steps of Video Composition to integrate all the materials (Section 4.3). Finally, we introduce the use of AnimeGANv2 for style transfer (Section 4.4).

### 4.1 Caption Generation

LLMs have powerful text understanding and generation capabilities. We need to use the generation capabilities of LLMs to process the input of intelligent directors, including images, videos and user requirements, and then generate high-quality video captions. However, LLMs can only process text and cannot handle other modal inputs, such as images and videos. Therefore, we use the multi-modal model LENS to generate descriptions for images. LENS performs computer vision and visual reasoning tasks through a frozen LLM and a set of "vision modules". LENS leverages these vision modules to obtain tags, attributes, and captions, which are then input into LLM to generate a response for a given question. We take the question *"What is the image about?"* and an image as input and obtain a description of the image.

For the video in the input of the intelligent director, we adopt a simple pHash algorithm to recognize the key frames of the video. pHash is the fingerprint of a multimedia file, derived from various characteristics of its content. If the features are similar, pHash will be "close" to each other. Specifically, we use the pHash to calculate the similarity between video frames and divide the video into some small segments according to the threshold (similarity < 0.6). Each small segment includes a key frame, that is, the video is down-sampled into a number of discrete key frames. After obtaining the key frames of the video, we can use the LENS to provide a description for each key frame.

The descriptions generated by the LENS model are simple descriptions of the images, lacking consideration of the contextual information from all images and videos, which are far from being story-telling captions. On the other hand, the initial arrangement order for the images and videos input by the user often cannot constitute a coherent story. Consequently, the initial arrangement order of the descriptions corresponding to the images and video key frames also fails to form cohesive video captions. To address this challenge, we propose utilizing the powerful generation and reasoning capabilities of ChatGPT to generate high-quality and

---

[3]https://tachibanayoshino.github.io/AnimeGANv2/

coherent video captions. Specifically, ChatGPT will employ its generation capability to expand descriptions and produce high-quality captions describing each scene. Additionally, ChatGPT will leverage its reasoning capability to analyze the relationships between the generated captions and rearrange their sequence to form a coherent story.

We integrate user requirements through careful prompt design to ensure that the generated video captions can form a coherent story and facilitate subsequent music retrieval and video composition. Users can input requirements to control the *{theme, time, location, requirement}* of the story. We design the prompt as:

**prompt = task description + input descriptions + detailed requirements**.

Among them, *task description* is a brief description of the task and specifies the topic, time and location (can be empty). The design of *task description* is as follows:

> **task description**: " I have a collection of photos and videos, but their order is chaotic. I hope you can help me arrange these materials in a certain order to create a video centered around the theme {theme}. Additionally, I'd like you to provide a smoothly written script that connects these images and videos into a cohesive story. The photos and videos were taken at {location}. They were captured at {time} I will provide descriptions for each image or video to give you an understanding of their content. "

*input description* are the captions generated by LENS in the previous Caption Generation. *input description* follows the following format:

> **input description**:
>
> > "Image 1: a lake in the middle of a forest with a mountain ...
> > Image 2: a stream running through a lush green forest
> > Image 3: a waterfall in the middle of a lush green forest
> >
> > $\vdots$
> >
> > Video 9: key frame 1: a large body of water surrounded ..."

*detailed requirements* are the detailed requirements of the task, including task splitting, output format, output language, word count, title recommendation, conclusion recommendation and music recommendation. The design of *detailed requirements* is as follows:

> **detailed requirements**: " I need you to do two things: (1) Rearrange the materials, grouping similar images together. If there's a clear timeline, arrange them in chronological order, otherwise, organize them based on your logical sequence.
> (2) Write a script according to the adjusted material sequence. I hope your script meets the following requirements: {requirement}. It should be concise, fluent, vivid, and the transitions between different materials should be natural. Each caption for the materials should not exceed 20 words. ... "

By structuring prompts in this way, we can leverage ChatGPT to generate high-quality and coherent video captions with recommended titles, conclusion, and music. Due to space limitations, readers are encouraged to refer to the supplementary materials for the complete prompt.

## 4.2 Music Retrieval

Music plays an irreplaceable role in video creation. Its selection is not only to fill gaps, but also to enhance the audience's perceptual experience. During the caption generation, we cleverly used prompts to ask ChatGPT to recommend a music name that match the video. We adopt a retrieval-based approach by searching a large music library to obtain the best-matched music to ensure that the selected music complements the overall presentation of the video.

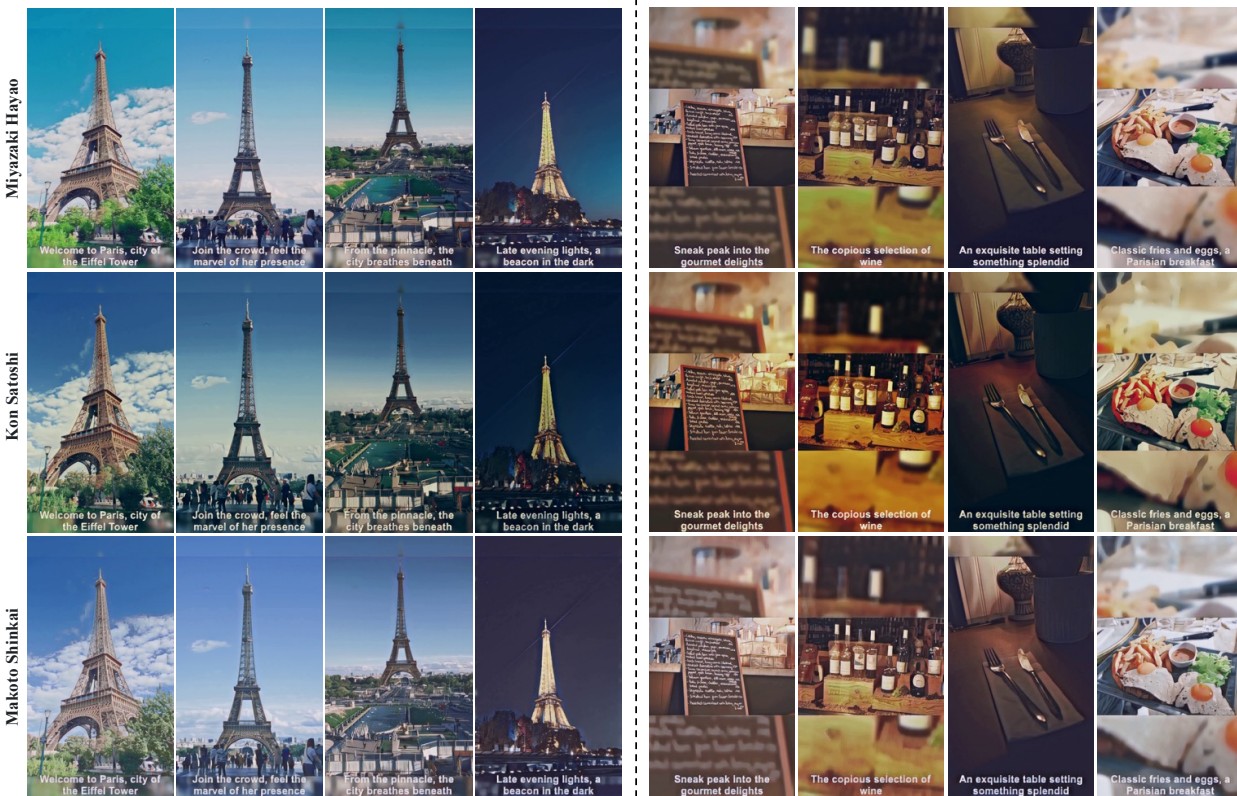

Figure 3: Results of style transfer with three animated styles on the Personal Album Dataset.

## 4.3 Video Composition

After acquiring materials for video creation, including images, videos, captions, and music, the next step is to combine all materials to create the synthesized video, referred to as video composition. We break down video composition into the following four steps:

**Caption Fusion** We first created a template for the opening and ending, inserted the title generated by ChatGPT into the opening, and inserted the conclusion into the ending. At the same time, insert the scripts generated by ChatGPT into the corresponding images or videos in order.

**Material Fine-tuning** To adapt the material to the target resolution of the synthesized video, we first resize the material to two different scales: one to match the height and the other to match the width. Then, if the width of the height-adapted material is smaller than the target width, we center the height-adapted material and use the width-adapted material, after applying Gaussian blur, as the background to fill the entire frame. Conversely, if the height of the width-adapted material is smaller than the target height, we center the width-adapted material and use the height-adapted material, after applying Gaussian blur, as the background to fill the frame. This ensures that the entire video frame is covered by the material, avoiding black borders.

**Switching Animation** The switching animation between materials in the video provides visual transitions and maintains coherence. We create four switching animations for material switching, including crossfade in, crossfade out, upward translation, and lateral translation. Random selection is made for each material switching.

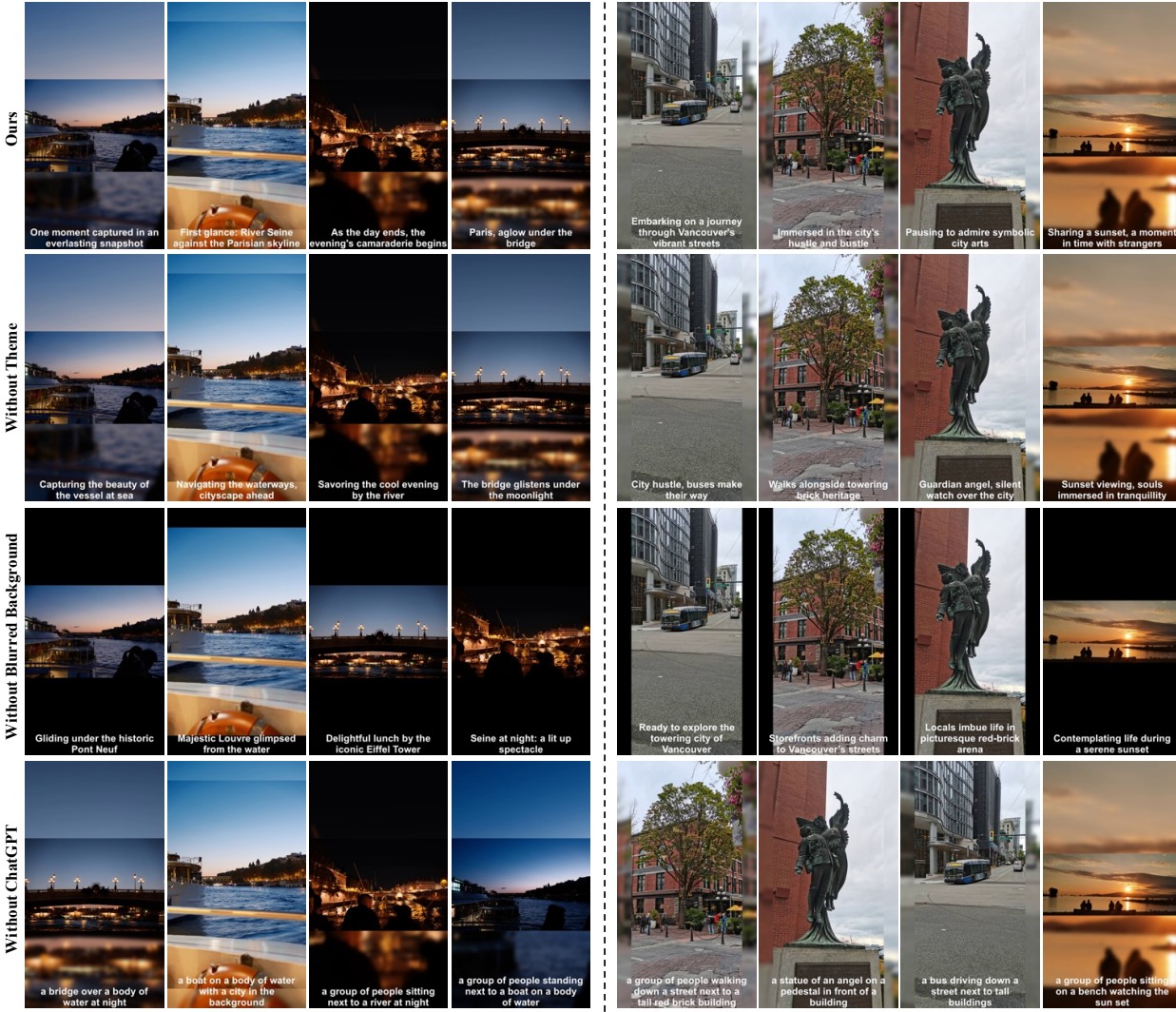

Figure 4: Comparison results between our framework and three baseline models on the Personal Album Dataset.

**Music Fusion**   During music fusion, standard beat detection (Ellis, 2007) is initially performed, followed by fine-tuning the duration of each material based on the music's rhythm. Specifically, setting the duration of the image defaults to four seconds. To align each material with the beat of the music when switching, the beat moment closest to the end time of the material will be identified as the new end time. Then, fine-tune the duration of the image or video to match the beat. Additionally, a minimum duration is set for each image or video clip to prevent flickering during video playback. This carefully designed process is designed to ensure seamless integration of music and video.

## 4.4   Style Transfer

Video style transfer empowers users with creative freedom by transforming the style of a video, making the content more aligned with specific themes. We utilize AnimeGANv2 (Chen & Liu, 2021) for style transfer on the generated videos, offering support for three animated styles: Miyazaki Hayao, Makoto Shinkai, and Kon Satoshi. Firstly, we extract the audio from the video, decompose the input video into individual frames, and subsequently apply AnimeGANv2 for style transfer on each video frame. Finally, we merge the audio with the style-transferred video frames to create a new video.

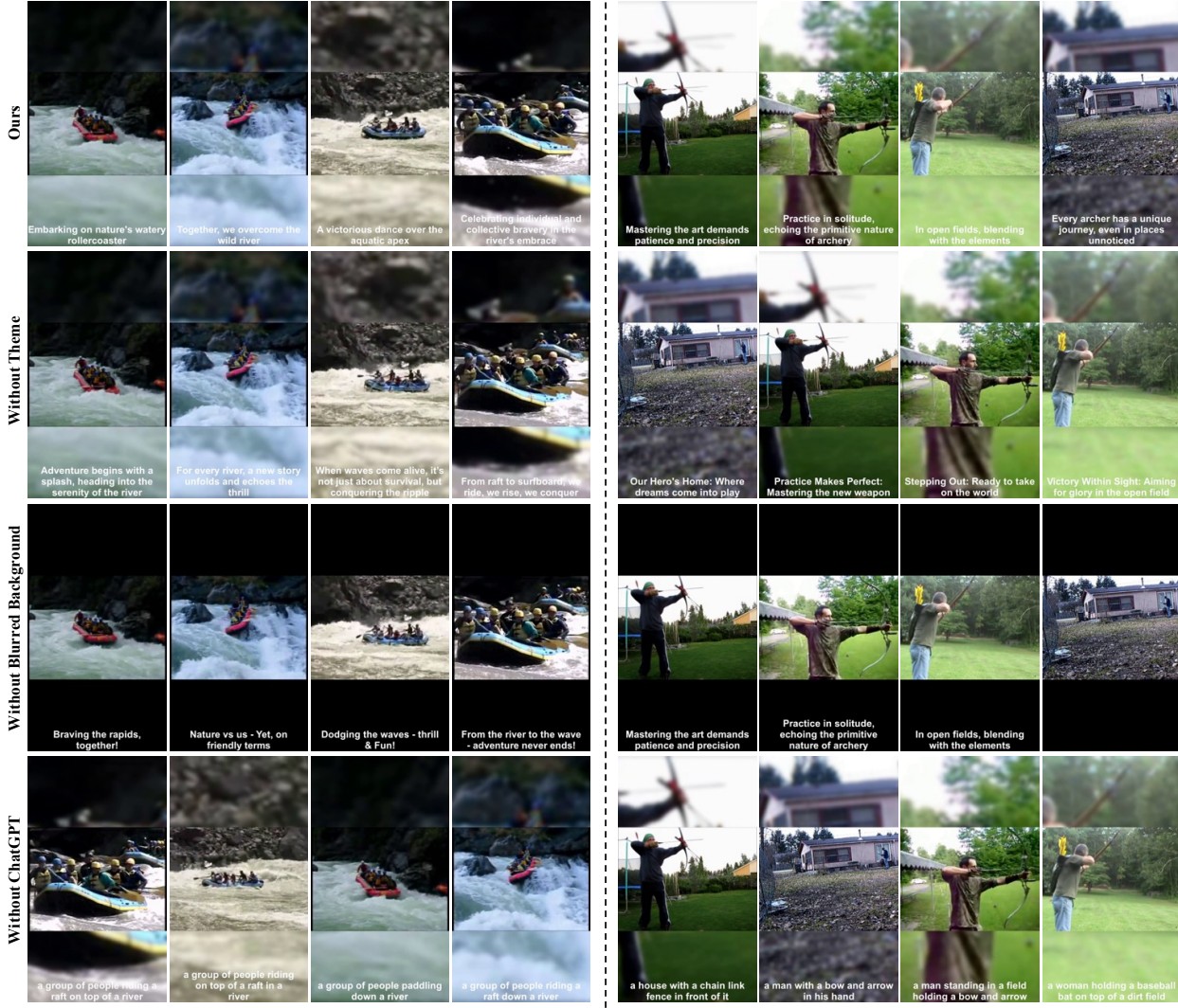

Figure 5: Comparison results between our framework and three baseline models on the UCF101-DVC Dataset.

# 5 Experiments

## 5.1 Datasets

The DVC task has not been specifically studied before, so we construct UCF101-DVC Dataset and Personal Album Dataset to verify the effectiveness of our proposed framework.

**UCF101-DVC Dataset**  UCF101 dataset (Soomro et al., 2012) consists of 13,320 video clips, which are classified into 101 categories. We randomly sampled 8 video clips from each class to form a video set, resulting in a total of 101 video sets, encompassing 808 video clips.

**Personal Album Dataset**  This dataset consists of photos and videos captured by five volunteers using their smartphones or cameras, totaling 25 sets of data, including 195 photos and 8 videos. The number of photos and videos in each set varies, with a maximum of 13 photos and 3 videos. Each set of data was captured at a specific location or centered around a particular theme, annotated by the volunteers.

Table 1: Quantitative comparison between our framework and three baseline models on TTR.

| Dataset | w/o Theme | w/o Blurred BG. | w/o ChatGPT | Ours |
|---------|-----------|-----------------|-------------|------|
| UCF101-DVC | **0.813** | 0.798 | 0.259 | 0.805 |
| PAD | **0.824** | 0.790 | 0.440 | 0.820 |

## 5.2 Evaluation Metrics

**TTR**  We use the Type-Token Ratio (TTR) Hess et al. (1984) to evaluate the lexical diversity of the generated video scripts. TTR measures the lexical diversity by calculating the ratio of the number of different words to the total number of words in a text. We compute the TTR for each video caption individually, and then average these values to obtain the overall mean TTR of all the videos. A higher TTR indicates increased lexical diversity in the text. In our study, analyzing the ratio of unique words to the total number of words in the generated video scripts allows us to quantify the level of lexical diversity in the scripts.

**GPT-4 Evaluator**  To comprehensively evaluate the performance of our framework, we introduce GPT-4 Achiam et al. (2023) as an evaluator. We design a specific evaluation prompt instructing GPT-4 to simulate an impartial judge in assessing the quality of text and visual elements of the synthesized video. Specifically, GPT-4 is tasked with scoring each aspect, including consistency, logicality, vividness, and overall effectiveness of the text and visuals, on a scale from 1 to 5. Additionally, it is required to provide corresponding justifications for each aspect. The entire process aims to simulate a real evaluation scenario, validating the performance of our proposed framework in the Dynamic Visual Composition task.

## 5.3 Implementation details

In the caption generation, we utilize the image-match[4] package to invoke the pHash algorithm, setting the similarity threshold of $< 0.6$ to extract keyframes from videos. Subsequently, we employ the open-source code from LENS[5] to generate descriptions. In the music retrieval, we use 9 music libraries, including QQMusic, NetEase Cloud Music, Kugou Music, KuWo Music, QianQian Music, 5sing Music, Migu Music, JOOX Music, 1ting Music. In the video composition, we leverage the moviepy[6] package to perform various video operations. The target resolution ($H_T \times W_T$) for videos is set at $1280 \times 720$ (720P) or $1920 \times 1080$ (1080P). For music fusion, we employ the librosa[7] package to detect beats in the music. In the style transfer, we utilize the open-source code of AnimeGANv2[8] to perform style transfer on videos.

## 5.4 Baselines

To evaluate the effectiveness of our Intelligent Director framework, we construct three baseline models for comparison: 1) **Without Theme** (w/o Theme): The theme in user requirements significantly affects the quality of captions generated by ChatGPT. We thus remove the theme to analyze its impact on results. 2) **Without Blurred Background** (w/o Blurred BG.): Gaussian blur in Material Fine-tuning step of video composition was excluded to study its influence on overall visual aesthetics. 3) **Without ChatGPT** (w/o ChatGPT): Replacing ChatGPT with a method randomly ordering descriptions and utilizing them as captions to examine the contribution of ChatGPT to the overall quality of generated captions.

## 5.5 Qualitative Results

As shown in Figure 3, we present key frames extracted from the generated videos. The captions generated by our framework exhibit a strong coherence with the videos, weaving together a compelling story, while the visuals are further enhanced through style transfer into three distinct animation styles. As shown in Figures

---

[4]`https://github.com/rhsimplex/image-match`

[5]`https://github.com/ContextualAI/lens`

[6]`https://github.com/Zulko/moviepy`

[7]`https://github.com/librosa/librosa`

[8]`https://github.com/TachibanaYoshino/AnimeGANv2`

Table 2: Quantitative comparison of GPT-4 scores between our framework and three baseline models on the PAD. The "Average" represents the average score across the four aspects. Larger numbers in the table indicate superior performance or preference.

| METHOD | w/o Theme | w/o Blurred BG. | w/o ChatGPT | Ours |
|---|---|---|---|---|
| Consistency | 4.68 | 4.60 | 4.36 | **4.72** |
| Logicality | 4.64 | 4.48 | 4.36 | **4.72** |
| Vividness | 4.40 | 4.32 | 4.04 | **4.52** |
| Overall | 4.48 | 4.44 | 4.28 | **4.56** |
| Average | 4.55 | 4.46 | 4.26 | **4.63** |

Table 3: Quantitative comparison of GPT-4 scores between our framework and three baseline models on the UCF101-DVC. The "Average" represents the average score across the four aspects. Larger numbers in the table indicate superior performance or preference.

| METHOD | w/o Theme | w/o Blurred BG. | w/o ChatGPT | Ours |
|---|---|---|---|---|
| Consistency | 4.25 | 4.31 | 3.34 | **4.50** |
| Logicality | 4.20 | 4.33 | 2.90 | **4.48** |
| Vividness | 3.66 | 3.71 | 2.58 | **4.03** |
| Overall | 3.64 | 3.69 | 2.70 | **3.93** |
| Average | 3.94 | 4.01 | 2.88 | **4.24** |

4 and 5, the results of "Without Theme" indicate that user-input themes can enhance the alignment between captions and visual effects of the videos. Additionally, the results of "Without Blurred Background" suggest that incorporating a Gaussian-blurred background can significantly improve the aesthetics of the videos. Finally, the results of "Without ChatGPT" demonstrate the pivotal role of ChatGPT in generating captions that are both coherent and storytelling.

## 5.6 Quantitative Results

**TTR** We present the TTR metric in Table 1 to compare the lexical diversity of our framework with three baselines. Notably, our framework, incorporating ChatGPT, demonstrates significant superiority over the baseline without ChatGPT. This discrepancy suggests that the text generated by the LENS model exhibits a noticeable amount of repetition. In DVC, such repetitive and low-quality text can detrimentally affect the user experience. Interestingly, the baseline without a theme (w/o Theme) outperforms our framework by a small margin in both datasets. This result suggests that a theme-centered script tends to concentrate more on a central concept, which may reduce diversity due to the focused narrative. However, it is worth noting that this trade-off in diversity may be beneficial for the overall coherence and alignment of the story, which is reflected in other qualitative metrics discussed later. On the other hand, removing the blurred background (w/o Blurred BG.) has a lesser impact on TTR compared to other baselines, indicating that background adjustments primarily influence visual aesthetics rather than text generation quality. The results indicate that each component contributes differently to the overall performance, with ChatGPT having the most significant effect on enhancing text diversity.

**GPT-4 Evaluator** We present a comparison of GPT-4 scores between our framework and three baselines across four aspects: consistency, logicality, vividness, and overall quality in Table 2 and Table 3. Our framework consistently achieves higher scores across all four aspects, demonstrating superior performance compared to the baselines. Specifically, the inclusion of ChatGPT notably improves logicality and vividness, highlighting its effectiveness in enhancing narrative coherence and engagement. Removing the theme or blurring backgrounds has less impact but still affects the overall quality. These results indicate that each component of our framework plays a crucial role in producing high-quality, coherent, and vivid storytelling videos.

Table 4: User Study Results. In this table, larger numbers indicate better performance or preference.

| METHOD | w/o Theme | w/o Blurred BG. | w/o ChatGPT | Ours |
|---|---|---|---|---|
| Correspondence | 3.71 | 3.74 | 3.44 | **3.94** |
| Coherence | 3.47 | 3.42 | 2.44 | **3.66** |
| Matching | 3.16 | 3.11 | 2.81 | **3.63** |
| Quality | 3.30 | 3.16 | 2.61 | **3.66** |

### 5.7 User Study

We conduct a comprehensive user study involving four methods, which include our Intelligent Director framework and three baseline variations derived from it. Each method synthesizes corresponding videos for all 25 sets of PAD data. A total of 15 participants take part in the user study, providing ratings based on four dimensions: 1) Correspondence: the alignment between the captions and the video; 2) Coherence: the coherence and vividness of the captions; 3) Matching: the matching between music and video; 4) Quality: the overall quality of the video. All evaluations are conducted using a Likert scale ranging from 1 to 5. As depicted in Table 4, our framework outperforms the three baseline models across all four evaluation metrics, indicating that our framework is preferred by users.

## 6 Limitations

While our Intelligent Director framework for Dynamic Visual Composition demonstrates promising results, there are several limitations that should be acknowledged. First, the reliance on ChatGPT and LENS introduces dependency on pre-trained models, which may limit the ability to adapt to domain-specific requirements. The quality of the generated videos is largely influenced by the performance of these pre-trained models, and inaccuracies in text or image understanding could lead to suboptimal storytelling. Second, the current framework struggles with handling long and complex narratives effectively, as ChatGPT can produce inconsistent storylines when dealing with large amounts of media content. Third, the video composition process, including music synchronization and style transfer, may still lack the fine control of manual editing, which can affect the overall quality in professional applications. Future work will focus on addressing these limitations by incorporating more domain-specific fine-tuning, improving consistency in long narrative generation, and enhancing the control over video composition parameters.

## 7 Conclusions

In this paper, we introduce the Dynamic Visual Composition (DVC) task to meet the demand for intelligent and user-friendly video creation tools. This is an interesting and challenging task, aiming to automatically integrate various media elements based on user requirements to create storytelling videos. We propose the Intelligent Director framework, effectively addressing the DVC task through four steps: Caption Generation, Music Retrieval, Video Composition, and Style Transfer. By constructing the UCF101-DVC and Personal Album datasets and conducting quantitative comparisons and user studies, we validated the effectiveness of the Intelligent Director framework in solving the DVC task and showcased its significant potential. We hope that our proposed task and framework will contribute to practical video creation and stimulate further research in the field.

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

## A  Complete Prompt for ChatGPT

In the Caption Generation of Intelligent Director, we leverage the generation and inference capabilities of ChatGPT to generate high-quality and coherent video captions. We incorporate user requirements into the prompt design to ensure that the generated video captions construct a cohesive story, enabling seamless

music retrieval and video composition. Users have the flexibility to input specific requirements to control the *{theme, time, location, requirement}* of the story. Our prompt is structured as:

**prompt = task description + input descriptions + detailed requirements**.

Among them, *task description* is a brief description of the task and specifies the topic, time and location (can be empty). The design of *task description* is as follows:

> **task description**: " I have a collection of photos and videos, but their order is chaotic. I hope you can help me arrange these materials in a certain order to create a video centered around the theme {theme}. Additionally, I'd like you to provide a smoothly written script that connects these images and videos into a cohesive story. The photos and videos were taken at {location}. They were captured at {time} I will provide descriptions for each image or video to give you an understanding of their content. "

*input description* are the captions generated by LENS in the previous Caption Generation. *input description* follows the following format:

> **input description**: "
>
> > Image 1: a lake in the middle of a forest with a mountain ...
> >
> > Image 2: a stream running through a lush green forest
> >
> > Image 3: a waterfall in the middle of a lush green forest
> >
> > $\vdots$
> >
> > Video 9: key frame 1: a large body of water surrounded ...
>
> "

*detailed requirements* are the detailed requirements of the task, including task splitting, output format, output language, word count, title recommendation, conclusion recommendation and music recommendation. The design of *detailed requirements* is as follows:

> **detailed requirements**: " I need you to do two things: (1) Rearrange the materials, grouping similar images together. If there's a clear timeline, arrange them in chronological order, otherwise, organize them based on your logical sequence.
> (2) Write a script according to the adjusted material sequence. I hope your script meets the following requirements: {requirement}. It should be concise, fluent, vivid, and the transitions between different materials should be natural. Each caption for the materials should not exceed 20 words.
> Also, please recommend a piece of instrumental music that suits this video. Finally, you should first rearrange the materials and then write corresponding captions based on the rearranged sequence.
> Below is an example output format:
> Order: (A sequence of Arabic numbers separated by commas, indicating the adjusted order of materials in your script)
> Title: A title for the beginning of the video, not exceeding 5 words
> Materials: Content of the materials rearranged in order
> Captions: A specific Arabic number (indicating the corresponding section of the material): The specific content of the caption
> Closing: A closing statement at the end of the video, not exceeding 8 words
> Music Recommendation: (Only provide the name of the music, no other words) "

By organizing prompts in this way, we can utilize ChatGPT to generate video captions of superior quality and coherence, complete with recommended titles, conclusions, and music recommendations.

## B Complete Prompt for GPT-4 Evaluator

In the Caption Generation, we leverage the generative and reasoning capabilities of ChatGPT to produce high-quality and coherent video captions. To comprehensively evaluate the performance of our framework, we introduce GPT-4 as an evaluator. We design a specific evaluation prompt instructing GPT-4 to simulate an impartial judge in assessing the quality of text and visual elements of the synthesized video. Specifically, GPT-4 is tasked with scoring each aspect, including consistency, logicality, vividness, and overall effectiveness of the text and visuals, on a scale from 1 to 5. Additionally, it is required to provide corresponding justifications for each aspect. The entire process aims to simulate a real evaluation scenario, validating the performance of our proposed framework in the Dynamic Visual Composition task.

> **GPT-4 prompt**: " You are an impartial judge tasked with evaluating the quality of edited video based on textual and visual elements. Your assessment should consider the overall coherence, creativity, and effectiveness of the content. You will rate the quality of the output on multiple aspects such as Consistency of text and video, Logicality, Vividness, and Overall.
>
> Evaluate
>
> Aspects
>
> Consistency of text and video: Rate the consistency of text and video on how well the text aligns with the visuals in the video clip, according to the consistency between what is described in the text and what is presented visually. A score of 5 indicates complete alignment, while a score of 1 suggests significant inconsistency.
>
> Logicality: Evaluate the logical flow of the text, examining how it contributes to a cohesive and sensible storyline. A score of 5 indicates a text that is logically sound, while a score of 1 suggests a lack of coherence and logic.
>
> Vividness: Rate the vividness on how well the text brings the video to life and enhances the viewer's experience. A score of 5 indicates highly vivid text, while a score of 1 suggests a lack of vividness and engagement.
>
> Overall: Rate the overall assessment on how effectively the text and visuals work together to create a compelling and coherent story. A score of 5 indicates good integration, while a score of 1 suggests poor integration. Format Please rate the quality of the edited video by scoring it from 1 to 5 individually on each aspect.
> - 1: strongly disagree
> - 2: disagree
> - 3: neutral
> - 4: agree
> - 5: strongly agree
> Now, please output your scores and a short rationale below in a json format by filling in the placeholders in []:
>
> {
> "consistency of text and video": {
> "reason": "[your rationale]",
> "score": "[score from 1 to 5]"
> },
> "logicality": {
> "reason": "[your rationale]",
> "score": "[score from 1 to 5]"
> },
> "vividness": {
> "reason": "[your rationale]",
> "score": "[score from 1 to 5]"
> },
> "aesthetic": {
> "reason": "[your rationale]",
> "score": "[score from 1 to 5]"
> },

```
"overall": {
"reason": "[your rationale]",
"score": "[score from 1 to 5]"
}
}
```

Material

The following is provided for your evaluation: an edited video, encompassing both the script within the video and a series of video frames.

Text Script:
{text}

Video Frames:
Video Frames are shown below.    ”

## C   Additional Qualitative Results

As shown in Figure 6, we present the results of style transfer of our framework on the PAD across three animated styles. At the same time, illustrated in Figure 7 are the results of style transfer of our framework on the UCF101-DVC Dataset across three animated styles. These results demonstrate the capability of our framework to generate diverse, aesthetically pleasing, and coherent storytelling videos. Furthermore, Figure 8 provides a comparative analysis between our framework and three baseline models on PAD, while Figure 9 provides the corresponding comparisons on the UCF101-DVC dataset. The results indicate that our framework produces captions closely aligned with the video content, forming a coherent story. The results of "Without Theme" suggest that user-provided themes contribute to captions better aligning with the visual content of the video. The results of "Without Blurred Background" highlight the significant enhancement of video aesthetics with the addition of Gaussian-blurred backgrounds. Lastly, the results of "Without ChatGPT" demonstrate the role of ChatGPT in generating more coherent and storytelling captions. These results offer a comprehensive view of the qualitative aspects of our proposed framework, further supporting its efficacy in Dynamic Visual Composition tasks.

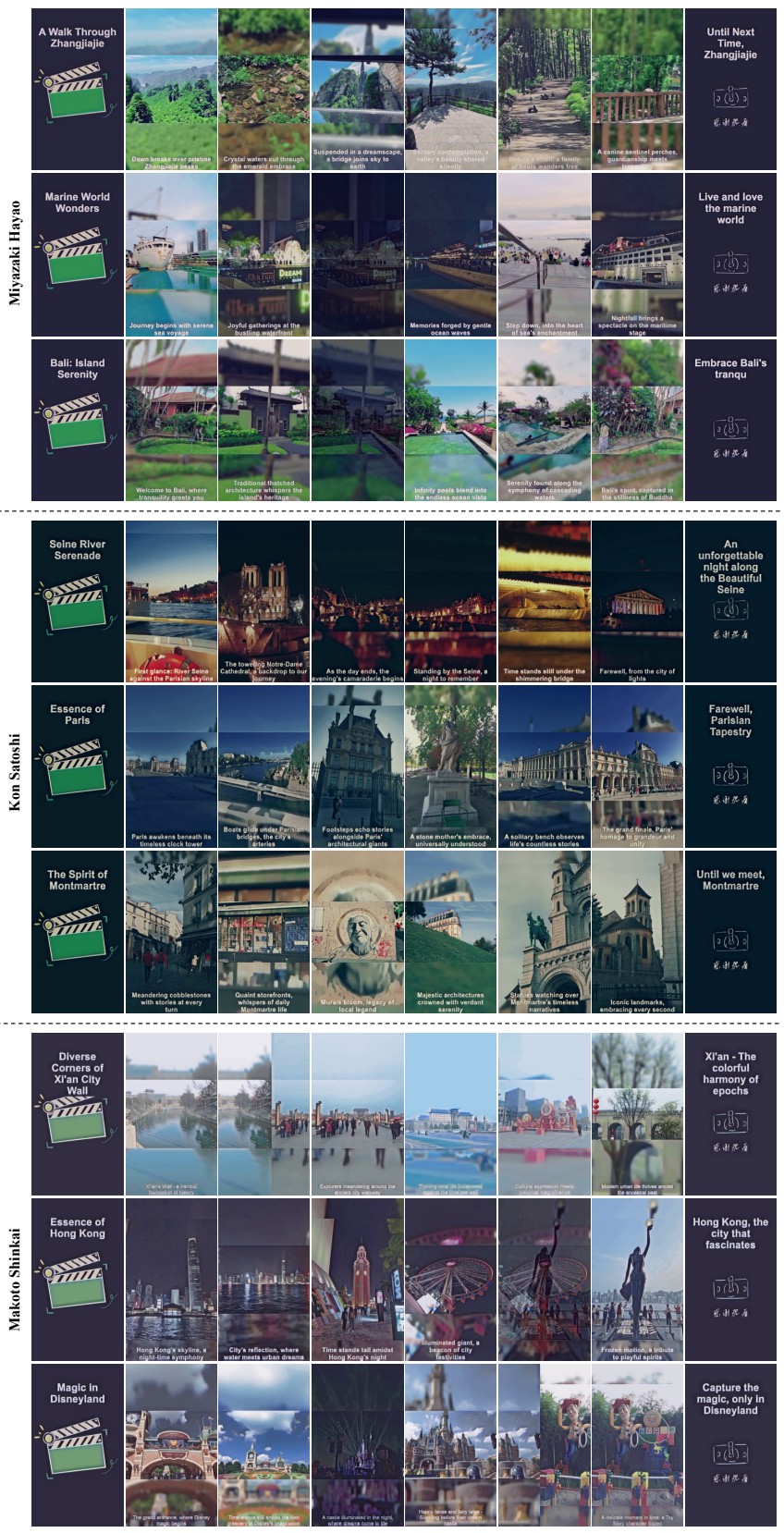

Figure 6: Results of style transfer with three animated styles on the Personal Album Dataset.

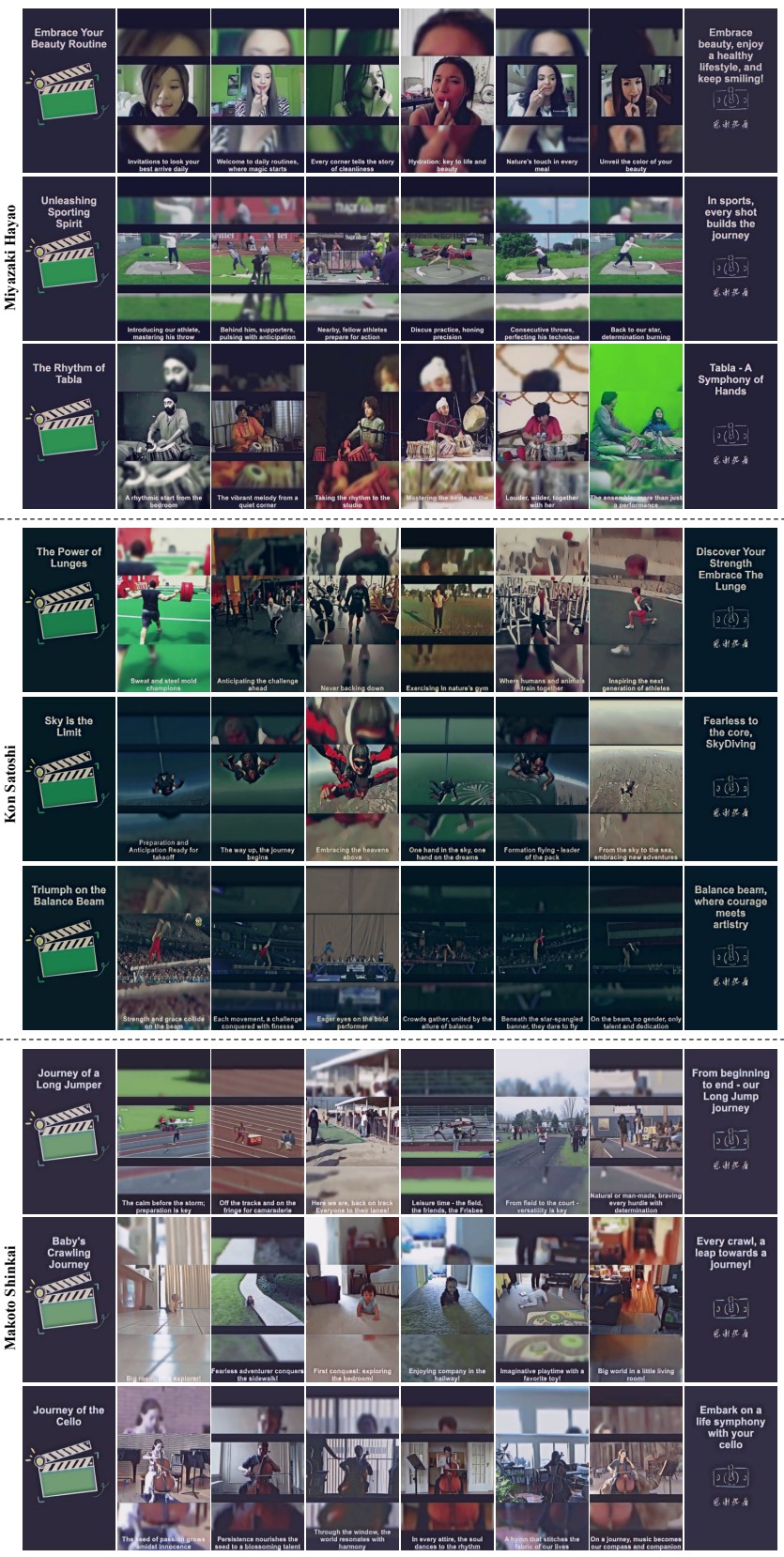

Figure 7: Results of style transfer with three animated styles on the UCF101-DVC Dataset.

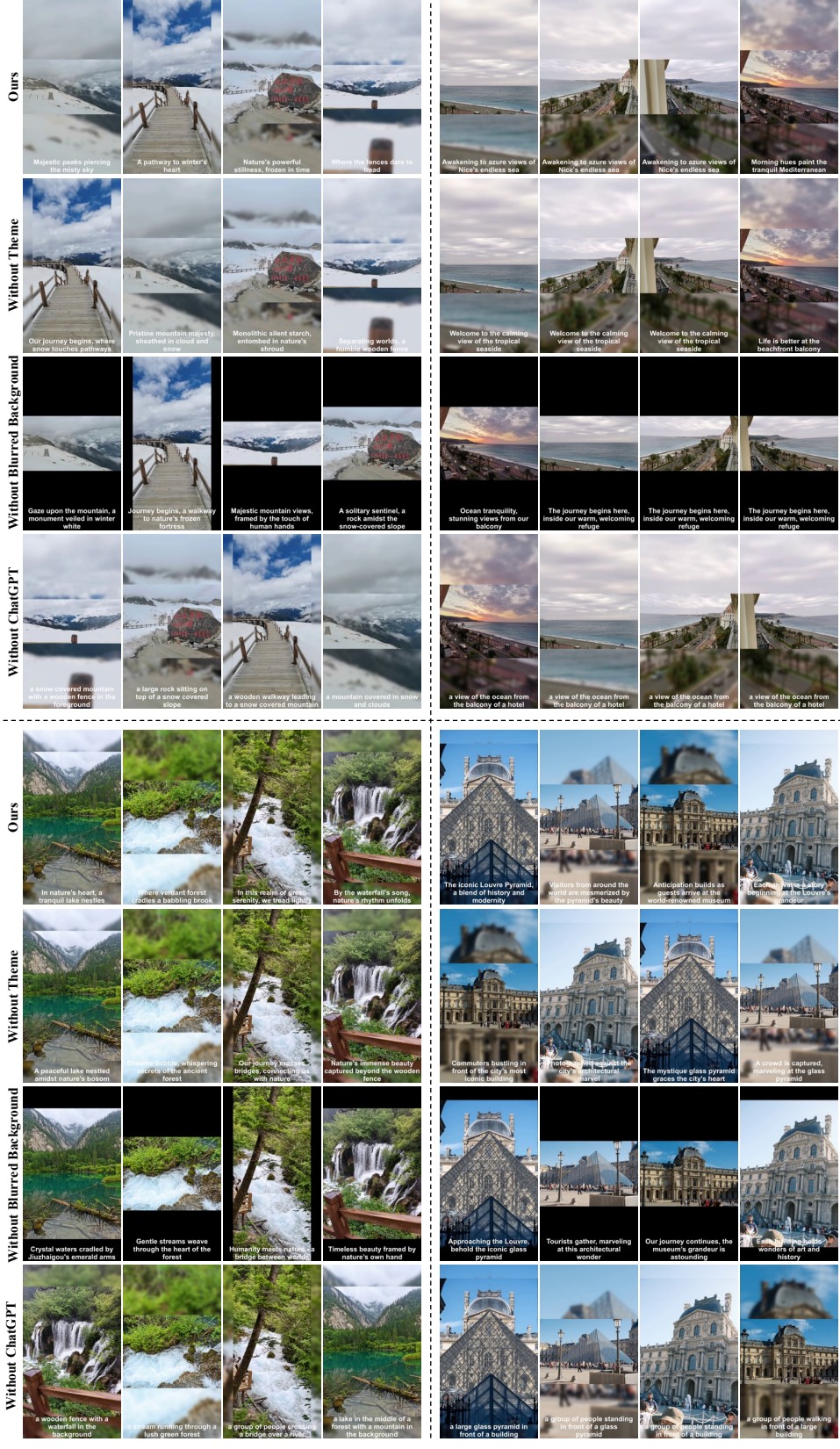

Figure 8: Comparison results between our framework and three baseline models on the Personal Album Dataset.

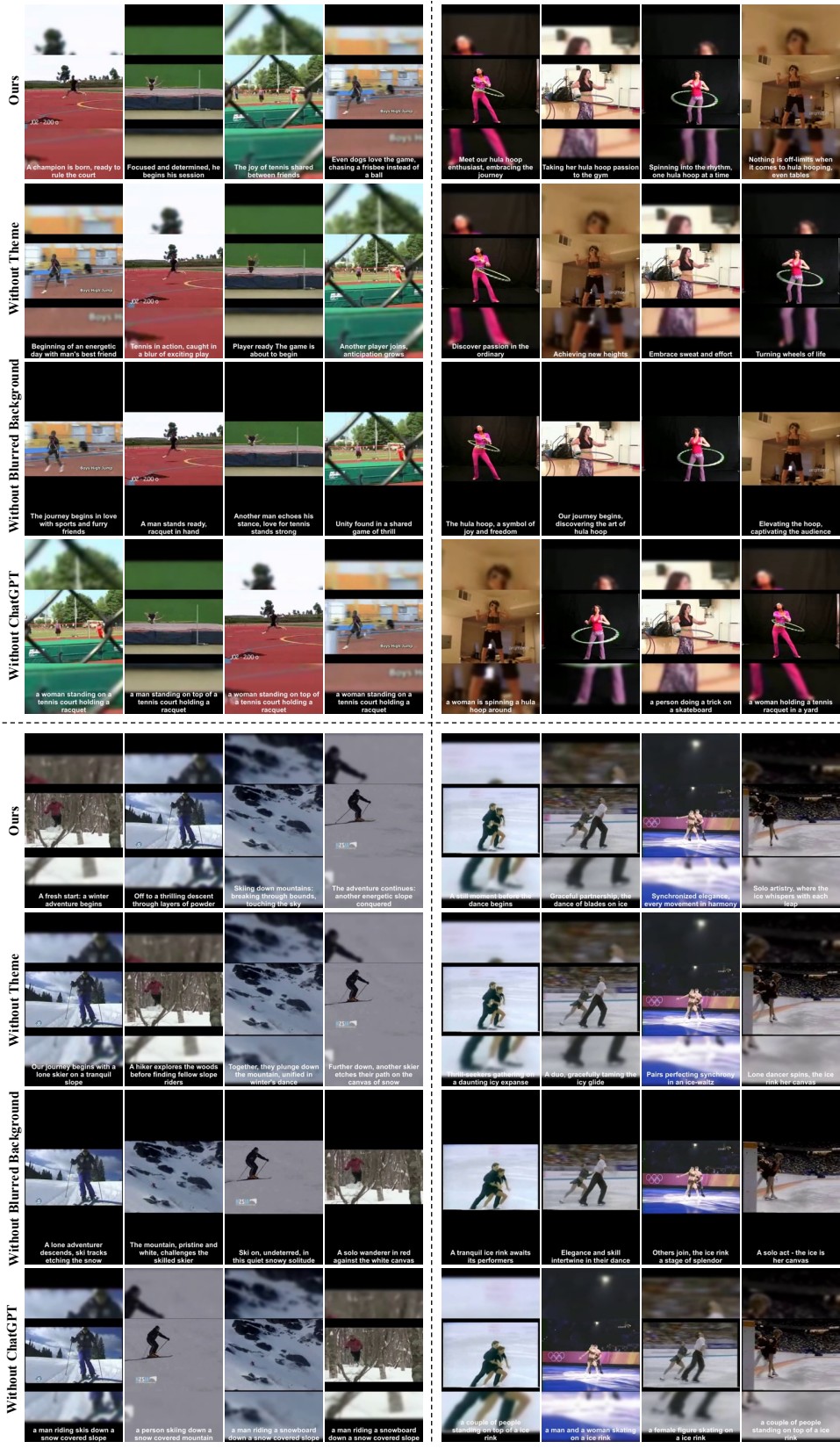

Figure 9: Comparison results between our framework and three baseline models on the UCF101-DVC Dataset.

