# OpenReview forum: "Intelligent Director: An Automatic Framework for Dynamic Visual Composition using ChatGPT"
_TMLR — Rejected by TMLR_

### Review · Reviewer_CFJq · 2024-09-20

**Summary Of Contributions:**

The authors propose a multimodal framework to assist users of video-based platforms in enhancing their content. The proposed framework is capable of performing tasks such as Caption Generation, Music Retrieval, Video Composition, and Style Transfer. It achieves these tasks by utilizing a series of pre-existing modules, with ChatGPT serving as the orchestrator. The framework is appealing from both a marketing and a scientific perspective. To demonstrate its scientific value, the authors propose a custom task called UCF101-DVC and discuss the proposed framework's performance, DVC.

Although the work is very interesting and the scientific methods are well introduced, the contribution is very poor in writing and is not in a condition to be published.

**Audience:**

Yes

**Broader Impact Concerns:**

The contribution may have a marginal impact since, although it is a useful approach, it could be applied as a tool to improve the enjoyment of online content, which is good but does not change the current canons of the common world.

**Claims And Evidence:**

No

**Requested Changes:**

The contribution is very interesting, and I think it has potential. I advise the authors to:

- Revise the writing style by better exemplifying the research questions and findings or contributions;
- build a more solid discussion by including an ablation study, and I can make comparisons with competitors;
- discuss the findings and limitations in detail, trying to understand the impact the paper can have in the world of research and how it can help people improve their daily lives (obviously, if it is possible)

**Strengths And Weaknesses:**

Strengths:

- Solid idea that combines strategic functional modules such as LENS and ChatGPT.
- The experimentation is well described.
- The qualitative discussion of the experiments (example of the task and images in the appendix) is very clear (see weaknesses for other points).

Weaknesses:

-  In the writing style, there are many ambiguous and unclear parts (the research questions are not clear, and the contributions are written sequentially at the end of the introduction).
- The quantitative discussion of the experiments is very poor, and there is a lack of ablation studies (Tables 3 and 4 are just baselines).
- Lack of discussion on the limitations of the contribution.
- Errors and inaccuracies: for example, the authors often use the contracted form "Sec," which, although not incorrect, is not pleasant to read.

---

> ### Author Response · Authors · 2024-10-09
> **Rebuttal by Authors**
>
> Dear Reviewer CFJq,
>
> Thank you for your feedback.  Below are our responses to your comments and questions.
>
> **Writing Style and Research Questions**
>
> In the revised version, we have clarified the research questions to make our objectives more explicit.
>
> **Quantitative Discussion and Ablation Study**
>
> In the revised version, we have added a more detailed quantitative discussion in Section 5.6. Furthermore, we have included the results of the ablation studies in Tables 1, 2, and 3.
>
> w/o Theme: This ablation study compares the impact of including or excluding the user requirements' theme on the quality of the generated captions.
>
> w/o Blurred BG: This ablation study examines the influence of Gaussian blur in the material fine-tuning step of video composition on overall visual aesthetics.
>
> w/o ChatGPT: This ablation study assesses the effect of using ChatGPT on the overall quality of the generated captions.
>
> **Discussion of Limitations**
>
> In the revised version, we have added a discussion of the limitations in Section 6. This section now highlights the challenges and potential future directions to further improve our proposed framework.
>
> **Writing Errors and Use of Contracted Forms**
>
> In the revised version, we have replaced "Sec." with "Section" throughout the paper to improve formality and readability.
>
> Again, we appreciate your review of our work. We hope our reply answers your questions.

---

### Review · Reviewer_Movr · 2024-09-25

**Summary Of Contributions:**

The authors propose a video composition mechanism, by stitching together images and videos with caption generation and styling. To that end, the authors i) use captioning using LENS to get captions from images and video key frames, ii) make coherent caption flows and music suggestions using ChatGPT, and iii) add styling and ratio fitting. The authors perform comparisons by ablating on their proposed components, do a quantitative experiment and user studies to showcase their method's effectiveness.

**Audience:**

Yes

**Broader Impact Concerns:**

There is no such concern. This is a video composition method.

**Claims And Evidence:**

Yes

**Requested Changes:**

Can you do a time comparison for different strategies to see which takes the bulk of time? I think this comparison will open up the paper to more future works.

**Strengths And Weaknesses:**

### Strengths
- The work is easy to follow and workflow is clean
- The proposed method shows satisfactory results in the experiments.


### Weaknesses/Questions
- I think a bottleneck of this method lies the captioning using LENS and getting a more coherent caption using ChatGPT. Why can't we have a multimodal LLMs (such as VITA [1]),  doing the captioning task instead?
- Given the nature of the paper, a simpler description of Material Fine-Tuning would be better suited for the audience.
- What is the time cost for different parts of the method? Having a smaller time footprint is important for video composition tasks.


References:
1. Fu, Chaoyou, et al. "Vita: Towards open-source interactive omni multimodal llm." arXiv preprint arXiv:2408.05211 (2024).

---

> ### Author Response · Authors · 2024-10-09
> **Rebuttal by Authors**
>
> Dear Reviewer Movr,
>
> Thank you for your constructive feedback and the valuable suggestions.  Below are our responses to your comments and questions.
>
> **Bottleneck: Captioning Using LENS and ChatGPT**
>
> Currently, multimodal LLMs such as LLaVA and Qwen-VL face challenges in generating high-quality captions for multiple images or videos, particularly when dealing with mixed inputs (images and videos together). They also struggle to produce storytelling captions or recommend suitable music, especially when the input consists of many images and videos. Our approach, using LENS to generate individual captions for each image and video, followed by ChatGPT to analyze, refine, and create coherent storytelling captions, was chosen to overcome these limitations.
>
> **Material Fine-Tuning Simplification**
>
> We have revised Section 4.3 Material Fine-tuning as follows:
>
> To adapt the material to the target resolution of the synthesized video, we first resize the material to two different scales: one to match the height and the other to match the width. Then, if the width of the height-adapted material is smaller than the target width, we center the height-adapted material and use the width-adapted material, after applying Gaussian blur, as the background to fill the entire frame. Conversely, if the height of the width-adapted material is smaller than the target height, we center the width-adapted material and use the height-adapted material, after applying Gaussian blur, as the background to fill the frame. This ensures that the entire video frame is covered by the material, avoiding black borders.
>
> **Time Cost for Different Parts**
>
> We have conducted a time analysis of different components of our framework using an RTX 3090 GPU, and the average time costs are as follows (in seconds):
>
> | **Component** | | **Average Time (seconds)** |
> | --- | --- | --- |
> | Caption Generation | Description Generation | 18.68 |
> | | ChatGPT | 14.40 |
> | Music Retrieval | | 120.34 |
> | Video Composition | Four Steps | 11.27 |
> | | Rendering Video | 584.16 |
> |  Style Transfer | | 25.34 |
>
> In Music Retrieval, music needs to be retrieved and downloaded, so it is time-consuming. The rendering video takes the most time, primarily due to the limitations of the moviepy library, especially when writing opening and closing animations. Reducing the time for this step remains an area for future optimization.
>
> Again, we appreciate your review of our work. We hope our reply answers your questions.

---

### Review · Reviewer_noYV · 2024-09-27

**Summary Of Contributions:**

This paper proposes a new task dynamic video composition to construct  It provides a framework and evaluation datasets for this task. It shows that by combining recent LLMs and other multimodal input processing modules we can automate the video generation process.

**Audience:**

Yes

**Broader Impact Concerns:**

I don't have concerns about this.

**Claims And Evidence:**

No

**Requested Changes:**

1. Use more convincing evaluation metrics to evaluate the generation quality.
2. Show the performance gain is statistically significant.
3. Provide more experiment details.

**Strengths And Weaknesses:**

Strengths:

1. This paper proposes a task dynamic video composition, which is indeed a very interesting task that might have real-world impact when people construct their videos from different elements.

2. The authors construct new datasets for evaluation.

3. The authors propose a framework to complete this task, which outperforms baseline methods in the evaluation.

4. The authors recruit human evaluators to evaluate the generation quality.

Weaknesses:

1. The evaluation criteria are not convincing: there are three evaluations used in this paper: type-token ratio, GPT-4 as an evaluator, and human user study.

Typo-token ratio (TTR) in the authors' definition measures the ratio of unique words out of all generated text. It cannot really show the information diversity of generated scripts but can merely show how linguistically rich the generated texts are. For instance, when a script says 'the movie is good and we can also say it's great or even excellent', it will have higher TTR compared to 'the movie is excellent but the movie music was not excellent' even when it is less informative.

User study is not convincing. There are only 15 user participants in this study and their backgrounds are underspecified. It's questionable whether these users are representative of the general user community of this video composition task. Moreover, there is only a very small performance gain in the proposed method compared to others. Since the evaluator population is so small, it's questionable whether the gain is statistically significant.

Using GPT-4 as evaluator is not convincing. Large Language Models are not trustworthy in evaluation. LLMs are reported to be biased in their evaluation and have a preference for their own generated content. It is not entirely surprising that GPT-4 prefers ChatGPT-generated content.

2. The authors mention that LLMs can only process text input and that's why they propose a framework of modulated components to compose videos. Since multimodal models are coming out, do we still need this framework?

3. Experiments lack descriptions of technical details: for example, in section 4.2, how do you determine some music is the best match and what is the database you use when you mention "We adopt a retrieval-based approach by searching a large music library to obtain the best-matched music to ensure that the selected music complements the overall presentation of the video"?

---

> ### Author Response · Authors · 2024-10-09
> **Rebuttal by Authors**
>
> Dear Reviewer noYV,
>
> Thank you for your detailed feedback. We have address your suggestions as follows:
>
> **Typo-Token Ratio**
>
> We agree with the reviewer that TTR is a metric for measuring lexical diversity rather than information diversity. We clarify this distinction in the revised version.
>
> **User Study Scale and Significance**
>
> Regarding the user study, we conducted evaluations using 25 samples with 15 participants, resulting in 25 * 15 = 375 individual trials. The number of trials exceeds related works, such as AutoStory (10 * 32 = 320), Make-A-Storyboard (4 * 20 = 80), and StoryGPT-V (100 * 3 = 300). Therefore, we believe our user study is sufficiently large to evaluate the performance of Intelligent Director compared to baseline models. Additionally, from the Average results in Table 3, our framework demonstrates significant improvements, such as a 5.7% gain compared to "w/o Blurred BG" and a 47.2% increase compared to "w/o ChatGPT," clearly showing that our method outperforms others in multiple aspects.
>
> **GPT-4 as an Evaluator**
>
> Recent studies show that LLMs like GPT-4 can achieve high agreement with human evaluators, making them reliable substitutes for human evaluators. Specifically:Hsu et al. (2023) showed GPT-4 as an effective evaluator for scientific figure captions. Sottana et al. (2023) demonstrated the reliability of GPT-4 as an evaluator for various sequence-to-sequence tasks. Wu et al. (2024) presented evidence that GPT-4V aligns well with human judgment in text-to-3D generation evaluations.
> These studies provide a basis for employing GPT-4 to evaluate video outputs' consistency, logicality, vividness, and overall effectiveness.
>
> [1] Hsu, Ting-Yao, et al. "Gpt-4 as an effective zero-shot evaluator for scientific figure captions." arXiv preprint arXiv:2310.15405 (2023).
>
> [2] Sottana, Andrea, et al. "Evaluation metrics in the era of GPT-4: reliably evaluating large language models on sequence to sequence tasks." arXiv preprint arXiv:2310.13800 (2023).
>
> [3] Wu, Tong, et al. "Gpt-4v (ision) is a human-aligned evaluator for text-to-3d generation." Proceedings of the IEEE/CVF Conference on Computer Vision and Pattern Recognition. 2024.
>
> **Necessity of the Proposed Framework**
>
> Regarding the necessity of our framework in light of emerging multimodal models: current multimodal models like LLaVA and Qwen-VL face challenges when generating a large number of captions of high quality across multiple images, videos, or mixed modalities. They also struggle to generate storytelling captions and recommend suitable music, particularly for extensive input content.
>
> **Experimental Details**
>
> In Section 5.3, we mentioned the use of 9 music libraries, including QQMusic, NetEase Cloud Music, Kugou Music, KuWo Music, QianQian Music, 5sing Music, Migu Music, JOOX Music, and 1ting Music. In music retrieval, we utilize the search interface of each library, which ranks the results according to relevance, and we select the top-ranked results as the retrieved music.
>
> Again, we appreciate your review of our work. We hope our reply answers your questions.

---

### Review · Reviewer_YJ4W · 2024-09-30

**Summary Of Contributions:**

This paper proposes a holistic, hierarchical approach to video generation, with a specific emphasis on narrative and multi-modal content.

**Audience:**

Yes

**Broader Impact Concerns:**

As with any creative generative AI application, this work has the potential for a broader impact on the creative industry, influencing creative individuals. Placing the work in context by considering how it might affect key stakeholders—such as video consumers, amateur creators, and professional video producers—would help address these broader implications.

**Claims And Evidence:**

No

**Requested Changes:**

Although evaluating a multi-faceted system like this one is inherently challenging, it is crucial to improve the scientific rigor in the paper’s experiments and evaluation to fully appreciate the work and highlight its value. Therefore, I recommend addressing the following critical points for acceptance:

1) Justifying the design decisions: Providing clear justifications for the design choices in the framework would significantly enhance its validity. Mentioning some ideas that were tried but did not work could offer valuable insights. Given the complexity of such a multi-faceted system, and the numerous potential design approaches, it is important for the community to understand why certain decisions made in this framework are preferable over others in certain respects.

2) Benchmarking and broader contextualization: While other studies and systems mentioned in the related work section do not directly address the DVC or, more broadly, the ID task, benchmarking the system’s subcomponents (e.g., captioning, music retrieval, video composition, style transfer) against existing literature would help ground the evaluation. This would strengthen both the baseline comparisons and the evaluation methodology. Additionally, situating the work within the broader field would better highlight its value. The current evaluation methodology, which relies on specific metrics like TTR and tools like the GPT-4 evaluator, feels too narrow. A more comprehensive set of metrics and a clearer rationale for their selection would provide a fuller understanding of the system’s performance.

**Strengths And Weaknesses:**

From the perspective of product development and the generated artifacts, the presented approach appears to have value. The thought process behind the development of this framework is evident, and there are useful insights in its design for other researchers and engineers in the field. The effectiveness of the design choices is demonstrated to some extent.

However, the main weakness of this paper is its lack of scientific rigor in experimentation and evaluation. Given the multi-faceted nature of the system, providing stronger support for the decisions made during the framework's development—arguably one of the paper’s key contributions—would enhance its validity. The rationale behind the evaluation methodology is unclear, particularly with respect to the selection of metrics and baselines. Further contextualizing the work within the broader field would also be beneficial.

---

> ### Author Response · Authors · 2024-10-09
> **Rebuttal by Authors**
>
> Dear Reviewer YJ4W,
>
> Thank you for your detailed feedback. We have address your suggestions as follows:
>
> **Justifying the Design Decisions**
>
> We initially experiment with using generative models, such as Stable Diffusion, to generate consistent and coherent video keyframes based on user requirements. Unfortunately, current generative models face significant challenges in generating consistent and coherent keyframes, often requiring extensive user intervention to adjust video scripts for a satisfactory output. This leads us to propose the DVC task, where users provide images, videos, and user requirements, and our Intelligent Director framework creates a storytelling video without requiring manual fine-tuning.
> Additionally, we attempt to use multi-modal large models, such as Qwen-VL, to generate captions for user-provided images and videos. We find that these models struggle to generate high-quality captions for a large number of images, multiple videos, or mixed inputs of images and videos. Moreover, they fail to create coherent storytelling captions and recommend suitable music. Therefore, our final approach uses LENS to generate captions for individual images and videos, followed by ChatGPT for further analysis and reasoning, ultimately producing coherent captions and recommending appropriate music.
>
> **Benchmarking and Broader Contextualization**
>
> Our primary focus is on the design of the overall framework rather than on evaluating individual subcomponents. Therefore, the paper emphasizes assessing the holistic performance of the framework rather than focusing on each subcomponent. Since the DVC task does not involve image or video generation but rather caption generation and video composition, we use TTR to evaluate the lexical diversity of the generated video captions. Furthermore, large language models (LLMs), such as GPT-4, achieve high agreement with human evaluators [1, 2, 3], making them reliable substitutes for human evaluators. As a result, we use GPT-4 as an evaluator to assess the consistency, logicality, vividness, and overall effectiveness of the generated videos. This choice allows us to provide a comprehensive evaluation of our framework's performance.
>
> [1] Hsu, Ting-Yao, et al. "Gpt-4 as an effective zero-shot evaluator for scientific figure captions." arXiv preprint arXiv:2310.15405 (2023).
>
> [2] Sottana, Andrea, et al. "Evaluation metrics in the era of GPT-4: reliably evaluating large language models on sequence to sequence tasks." arXiv preprint arXiv:2310.13800 (2023).
>
> [3] Wu, Tong, et al. "Gpt-4v (ision) is a human-aligned evaluator for text-to-3d generation." Proceedings of the IEEE/CVF Conference on Computer Vision and Pattern Recognition. 2024.
>
> Again, we appreciate your review of our work. We hope our reply answers your questions.

---

### Decision · Action_Editor_cVD8 · 2024-10-28

**Recommendation:** Reject

**Comment:**

Regarding the comments of the reviewers, the majority of them lean to reject this paper. The AC have read the paper, reviews, and the responses, and agree with the consensus made by the reviewers. Here are the summarized reasons for the recommendation: 1) Lack of Rigorous Experimental Design: The selection criteria for both the experimental and control groups are not clearly defined, which may lead to biased results. 2) Insufficient Sample Size: The small sample size limits the generalizability of the conclusions. 3) Weak Theoretical Foundation: The study lacks a thorough analysis of existing literature and theoretical support, making the interpretation of the results less compelling.

**Audience:**

The researchers who study on LLM-based deep learning will be interested.

**Claims And Evidence:**

This work provides both qualitative and quantitative experimental results to demonstrate the effectiveness of the framework. It includes metrics for evaluation, baseline comparisons, and ablation studies to show the contribution of each component.

**Resubmission Of Major Revision:**

The authors may consider submitting a major revision at a later time.